# Adaptive Gradient Methods with Local Guarantees

**Zhou Lu**[1,2] **Wenhan Xia**[1,2] **Sanjeev Arora**[1] **Elad Hazan**[1,2]

[1]Princeton University
[2]Google AI Princeton

**Abstract** Adaptive gradient methods are the method of choice for optimization in machine learning and used to train the largest deep models. In this paper we study the problem of learning a local preconditioner, that can change as the data is changing along the optimization trajectory. We propose an adaptive gradient method that has provable adaptive regret guarantees vs. the best local preconditioner. To derive this guarantee, we prove a new adaptive regret bound in online learning that improves upon previous adaptive online learning methods.

We demonstrate the robustness of our method in automatically choosing the optimal learning rate schedule for popular benchmarking tasks in vision and language domains. Without the need to manually tune a learning rate schedule, our method can, in a single run, achieve comparable and stable task accuracy as a fine-tuned optimizer.

## 1 Introduction

Adaptive gradient methods have revolutionized optimization for machine learning and are routinely used for training deep neural networks. These algorithms are stochastic gradient based methods, incorporating a changing data-dependent preconditioner. The common intuitive understanding of their success is their ability to change the preconditioner, or learning rate matrix, per coordinate and on the fly. A methodological way of changing the learning rate allows treating important coordinates differently as opposed to commonly appearing features of the data, and thus achieve faster convergence.

In this paper we investigate whether a more refined goal can be obtained: namely, can we adapt the learning rate per coordinate, and also in short time intervals? The intuition guiding this search is the rising popularity in "exotic learning rate schedules" for training deep neural networks. The hope is that an adaptive learning rate algorithm can automatically tune its preconditioner, on a per-coordinate and per-time basis, such to guarantee optimal behavior even locally.

To pursue this goal, we use and improve upon techniques from the literature on adaptive regret in online learning to create a provable method that is capable of attaining optimal regret in any subinterval of the optimization trajectory. We then test the resulting method and compare it to learning a learning rate schedule from scratch.

### 1.1 Informal statement of our results

The (stochastic/sub)-gradient descent algorithm is given by the following iterative update rule:

$$x_{\tau+1} = x_\tau - \eta_\tau \nabla_\tau,$$

where $x_\tau$ are the parameters to be optimized, $\nabla_\tau$ is a random variable whose expectation is the (sub)gradient of the objective, and $\eta_\tau$ is the learning rate parameter, which can be either a scalar or a matrix. If $\eta_\tau$ is a matrix, it is usually called a preconditioner. A notable example for a preconditioner is when $\eta_\tau$ is equal to the inverse Hessian (or second differential), which gives Newton's method.

In a nutshell, the convergence rate of adaptive gradient methods can be explained as follows. Let $\nabla_1, ..., \nabla_T$ be the gradients observed in an optimization trajectory. It is well established that stochastic gradient descent (SGD) approaches optimality at a rate of $\sim \frac{1}{\sqrt{T}}$.

Adaptive gradient methods can improve upon this rate in a subtle way. The AdaGrad algorithm (and more general adaptive gradient methods) achieve the same guarantee at a rate of

$$\sim \frac{\sqrt{\min_{H \in \mathcal{H}} \sum_\tau \|\nabla_\tau\|_H^{*2}}}{T},$$

where $\mathcal{H}$ is a family of matrix norms, most commonly those with a bounded trace. Thus, adaptive gradient methods can improve upon the rate of vanilla SGD in certain geometric favorable scenarios.

In this paper we improve upon this guarantee in terms of the local performance over any interval of the optimization trajectory. This allows the algorithm to change learning rate in any interval, and over any coordinate. The formal guarantee is given below ($\{I_j\}$ is a partition of $[1, T]$):

**Theorem 1.** *The convergence rate of Algorithm 1 can be upper bounded by:*

$$\sim \tilde{O}\left(\frac{\min_k \min_{H_1,\ldots,H_k \in \mathcal{H}} \sum_{j=1}^k \sqrt{\sum_{\tau \in I_j} \|\nabla_\tau\|_{H_j}^{*2}}}{T}\right)$$

The convergence results above is derived using the methodology of regret in online convex optimization (OCO). Our main technical contribution is a strengthening of the best adaptive regret bound for general OCO:

**Theorem 2.** *The regret of Algorithm 1 can be upper bounded by:*

$$\tilde{O}\left(\min_k \min_{H_1,\ldots,H_k \in \mathcal{H}} \sum_{j=1}^k \sqrt{\sum_{\tau \in I_j} \|\nabla_\tau\|_{H_j}^{*2}}\right)$$

Our new upper bound improves over all previous results in terms of adaptive regret. For regret over any interval $I = [s, t]$, Hazan and Seshadhri (2007) first achieved regret $\tilde{O}(\sqrt{T})$, which was later improved by Daniely et al. (2015), Jun et al. (2017) to $\tilde{O}(\sqrt{|I|})$. Recently Cutkosky (2020) achieved the bound $\tilde{O}(\sqrt{\sum_{\tau=s}^t \|\nabla_\tau\|^2})$, and our result further improves it to $\tilde{O}(\min_{H \in \mathcal{H}} \sqrt{\sum_{\tau=s}^t \|\nabla_\tau\|_H^{*2}})$.

## 1.2 Setting and Preliminaries

We consider the problem of online convex optimization. At each round $\tau$, the learner outputs a point $x_\tau \in \mathcal{K}$ for some convex domain $\mathcal{K} \subset R^d$, then suffers a convex loss $\ell_\tau(x_\tau)$ which is chosen by the nature. In the most common setting, the learner gains access to the sub-gradients $\nabla_\tau$ of $\ell_\tau()$ at any $x_\tau$. The goal of the learner in OCO is to minimize regret, defined as

$$\text{Regret} = \sum_{\tau=1}^T \ell_\tau(x_\tau) - \min_{x \in \mathcal{K}} \sum_{\tau=1}^T \ell_\tau(x).$$

Similarly, we can define adaptive regret to measure the worst case regret over any interval.

$$\textbf{Adaptive-Regret} = \sup_{I=[s,t] \subset [1,T]} \sum_{\tau=s}^t \ell_\tau(x_\tau) - \min_{x^* \in X} \sum_{\tau=s}^t \ell_\tau(x^*).$$

Henceforth we make the following basic assumptions for simplicity:

**Assumption 1.** *There exists $D, D_\infty > 1$ such that $\|x\|_2 \leq D$ and $\|x\|_\infty \leq D_\infty$ for any $x \in \mathcal{K}$.*

**Assumption 2.** *There exists $G > 1$ such that $\|\nabla_\tau\|_2 \leq G, \forall \tau \in [1, T]$.*

We make the notation of the norm $\|\nabla\|_H$, for any PD matrix $H$ to be:

$$\|\nabla\|_H = \sqrt{\nabla^\top H \nabla}$$

The dual norm is defined as $\|\nabla\|_H^* = \sqrt{\nabla^\top H^{-1} \nabla}$. We denote $\mathcal{H} = \{H | H \succeq 0, tr(H) \leq d\}$.

**Algorithm 1** Strongly Adaptive Regret MUltiplicative-wEight-based AdaGrad (SAMUEL )

---

Input: OCO algorithm $\mathcal{A}$, interval set $S$, constant $Q = 4\log(dTD^2G^2)$.

Initialize: for each $I \in S$, $Q$ copies of OCO algorithm $\mathcal{A}_{I,q}$. Set $\eta_{I,q} = \frac{1}{2GD2^q}$ for $q \in [1, Q]$.

Initialize $w_1(I, q) = \min\{1/2, \eta_{I,q}\}$ if $I = [1, s]$, and $w_1(I, q) = 0$ otherwise for each $I \in S$.

**for** $\tau = 1, \ldots, T$ **do**

  Let $x_\tau(I, q) = \mathcal{A}_I(\tau)$, and $W_\tau = \sum_{I \in S(\tau), q} w_\tau(I, q)$.

  Predict $x_\tau = \sum_{I \in S(\tau), q} w_\tau(I, q) x_\tau(I, q) / W_\tau$.

  Receive loss $\ell_\tau(x_\tau)$, define $r_\tau(I) = \ell_\tau(x_\tau) - \ell_\tau(x_\tau(I, q))$.

  For each $I = [s, t] \in S$, update $w_{\tau+1}(I, q)$ as follows,

$$
w_{\tau+1}^{(I,q)} = \begin{cases}
0 & \tau + 1 \notin I \\
\min\{1/2, \eta_{I,q}\} & \tau + 1 = s \\
w_\tau(I, q)(1 + \eta_{I,q} r_\tau(I)) & \textbf{else}
\end{cases}
$$

**end for**

---

## 2 Algorithm and Main Theorem

At a high level, we choose a set of intervals $S$ and hold a blackbox algorithm instance $\mathcal{A}_I$ on each $I$ in the set, then run a variant of multiplicative weight algorithm on these blackbox algorithm instances. Our main technical contribution is to make copies of each 'expert' $\mathcal{A}_I$ with different learning rates in the multiplicative weight algorithm, so that one of them is guaranteed to be near optimal. Without loss of generality, we assume $T = 2^k$ and define the geometric covering intervals:

**Definition 3.** Define $S_i = \{[1, 2^i], [2^i + 1, 2^{i+1}], \ldots, [2^k - 2^i + 1, 2^k]\}$ for $0 \le i \le k$. Define $S = \cup_i S_i$ and $S(\tau) = \{I \in S | \tau \subset I\}$.

For $2^k < T < 2^{k+1}$, one can similarly define $S_i = \{[1, 2^i], [2^i + 1, 2^{i+1}], \ldots, [2^i \lfloor \frac{T-1}{2^i} \rfloor + 1, T]\}$, see Daniely et al. (2015). Hencefore at any time $\tau$ the number of 'active' (with positive weight) intervals is only $O(\log(T))$, this guarantees that the running time and memory cost per round of SAMUEL is as fast as $O(\log(T))$. It's worth to notice that $q$ doesn't affect the behavior of $\mathcal{A}_{I,q}$ and only takes affect in the multiplicative weight algorithm, and that $r_\tau(I, q)$ and $x_\tau(I, q)$ doesn't depend on $q$ so we may write $r_\tau(I)$ and $x_\tau(I)$ instead for simplicity.

We provide our main result, a full-matrix strongly adaptive regret bound for Algorithm 1.

**Theorem 2.1.** *Under assumptions 1 and 2, when Adagrad is used as the blackbox $\mathcal{A}$, the total regret Regret$(I)$ of the multiplicative weight algorithm in Algorithm 1 satisfies that for any interval $I = [s, t]$,*

$$
Regret(I) = O(D \log(T) \max\{G\sqrt{\log(T)}, d^{\frac{1}{2}} \sqrt{\min_{H \in \mathcal{H}} \sum_{\tau=s}^{t} \|\nabla_\tau\|_H^{*2}}\})
$$

The proof idea is seeing algorithms over different intervals as experts, then running a multiplicative weight algorithm on top of them is expected to achieve near-optimal regret over any of the intervals. Previous methods failed to achieve the optimal full-matrix bound, because it requires setting $\eta$ optimally in advance, however the optimal value depends on future gradients which we can't anticipate. We incorporate different $\eta$s to the experts of the internal MW to overcome this technical challenge.

## 3 Experiments

We demonstrate the effectiveness of our optimization method on popular vision and language benchmarks: image classification on CIFAR-10 and ImageNet, and sentiment classification on SST-2. On all tasks, SAMUEL stably achieve high accuracy without learning rate schedule tuning.

For experiments, we made a few adjustments to our theoretical algorithm 1 to be computationally efficient in practice. We take a fixed number of experts with exponential decay factor on the history as shown below. Additionally, we sample experts instead of taking convex combination of them.

$$x_{t+1} = x_t - \frac{\eta}{\sqrt{\epsilon I + \sum_{\tau=1}^{t} \alpha^{t-\tau} \nabla_\tau \nabla_\tau^\top}} \nabla_t$$

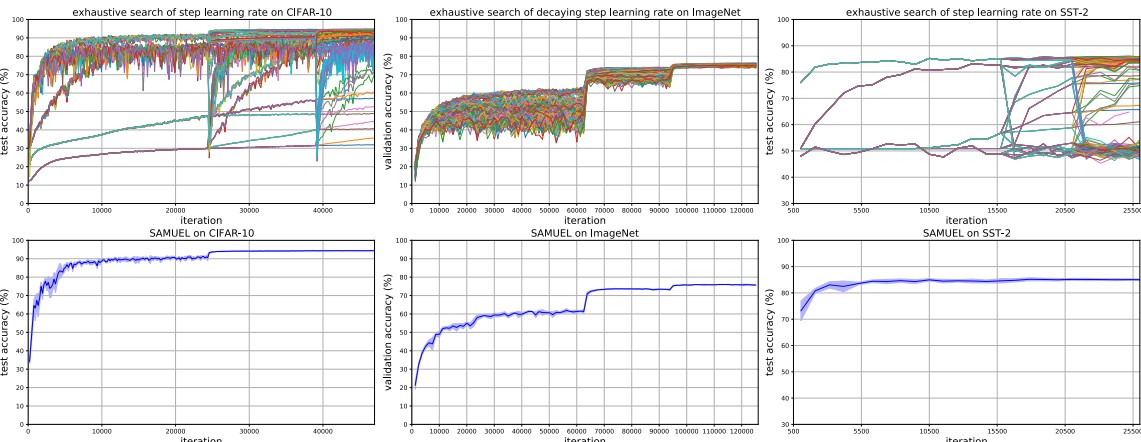

Figure 1: Comparison of exhaustive searched step learning rate schedule (top) and SAMUEL (bottom) on CIFAR-10, ImageNet and SST-2.

### 3.1 Vision tasks

**CIFAR-10 classification**: We compare a ResNet-18 model trained with SAMUEL to ResNet-18 trained with AdaGrad using brute-force searched learning rate schedules. We processed and augmented the data following He et al. (2016). All experiments were conducted on TPU-V2 hardware. For training, we used a batch size of 256 and 250 total epochs with a step learning rate schedule. We fixed the learning rate stepping point at epoch 125 and 200, and provided five possible candidate learning rates {0.0001, 0.001, 0.01, 0.1, 1} for each region. Thus an exhaustive search yielded 125 different schedules for the baseline AdaGrad method. For a fair comparison, we adopted the same learning rate changing point for our method.

We compared the test accuracy curves of the baselines and our methods in Fig.1. The left plot in Fig.1 displays 125 runs using AdaGrad for each learning rate schedule, where the highest accuracy is 94.95%. A single run of SAMUEL achieves 94.76% with the same random seed (average among 10 different random seeds is 94.50%), which ranks in the top 3 of 125 exhaustively searched schedules.

**ImageNet**: We continue examining the performance of SAMUEL on the large-scale ImageNet dataset. We trained ResNet-50 with exhaustive search of learning rate schedules and compare with SAMUEL. We also consider a more practical step learning rate scheduling scheme where the learning rate after each stepping point decays. Specifically, the candidate learning rates are {0.2, 0.4, 0.6, 0.8, 1.0} in the first phase and decay by 10× when stepping into the next phase. The total training epochs are 100 and the stepping position is set at epoch 50 and 75. We adopted the pipeline from Heek et al. (2020) for image pre-processing and model training. For both baselines and SAMUEL, we used the SGD optimizer with nesterov momentum of 0.9. All experiments were conducted on TPU-V2 hardware with training batch size of 1024.

The second column of Fig.1 displays the comparison of the exhaustive search baseline (top) to SAMUEL (bottom). The best validation accuracy out of exhaustively searched learning rate

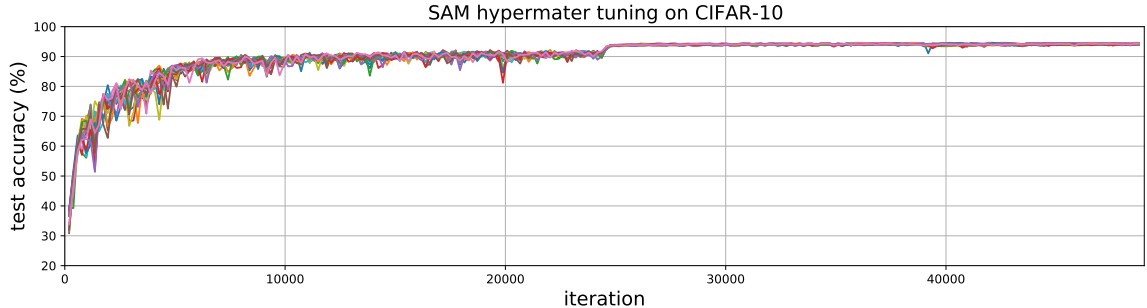

Figure 2: stability study of SAMUEL with different hyperparameters.

schedules is 76.32%. SAMUEL achieves 76.22% in a single run (average among 5 different random seeds is is 76.15%).

### 3.2 Language task

We consider tasks in the language domain and conducted experiments on the Stanford Sentiment Treebank SST-2 dataset. We used the pipeline from (Heek et al., 2020) for pre-processing the SST-2 dataset and trained a simple bi-directional LSTM text classifier. The total training epoch is 25 with stepping learning rate position at epoch 15 and 20. We used SGD with momentum of 0.9 and additive weight decay of 3e-6. The training batch size in both baseline and SAMUEL is 64. The learning rate schedule setting is the same as that of CIAR-10.

The right column of Fig. 1 shows that the best accuracy of exhaustive search is 86.12%, and the accuracy of SAMUEL using the same seed is 85.55% (average is 85.58% among 10 different random seeds), showing that our algorithm can achieve comparable performance not only on vision datasets but also on language tasks.

### 3.3 Stability of SAMUEL

We demonstrated the stability of SAMUEL with hyperparameter tuning. Since our algorithm will automatically selects the optimal learning rate, the only tunable hyperparameters are the number of $\eta$ and the number of history decaying factor $\alpha$. We conducted 18 trials with different hyperparameter combinations and display the test accuracy curves in Fig.2. Specifically, we considered the number of decaying factors $\alpha$ with values $\{2, 3, 6\}$ and the number of $\eta$ with values $\{5, 10, 15, 20, 25, 30\}$. As Fig.2 shows, all trials in SAMUEL converge to nearly the same final accuracy regardless of the exact hyperparameters.

## 4 Limitations and Broader Impact Statement

Our results address a more efficient optimization method for deep learning, that highlights resource expenditure that is many times neglected. For example, many research papers report accuracy or speedup **after hyperparameter optimization** (HPO). However, the hyperparameter sweep many times increases resource constraints by orders of magnitude.

Our experimental setup encourages researchers to address the "hidden" costs of HPO in mainstream optimization of deep learning. We believe this is a more correct way to measure efficiency and resource expenditure, and hope this will lead to increased awareness of the environmental toll of large model training.

In the short term, it is possible that our metric will be adopted, and lead to an *increase* in resource usage, as researchers will try to demonstrate improved efficiency taking into account the HPO process.

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
