# Adaptive Gradient Methods with Local Guarantees

## Abstract

Adaptive gradient methods are the method of choice for optimization in machine learning and used to train the largest deep models. In this paper we study the problem of learning a local preconditioner, that can change as the data is changing along the optimization trajectory. We propose an adaptive gradient method that has provable adaptive regret guarantees vs. the best local preconditioner. To derive this guarantee, we prove a new adaptive regret bound in online learning that improves upon previous adaptive online learning methods. We demonstrate the robustness of our method in automatically choosing the optimal learning rate schedule for popular benchmarking tasks in vision and language domains. Without the need to manually tune a learning rate schedule, our method can, in a single run, achieve comparable and stable task accuracy as a fine-tuned optimizer.

## 1 Introduction

Adaptive gradient methods have revolutionized optimization for machine learning and are routinely used for training deep neural networks. These algorithms are stochastic gradient based methods, that also incorporate a changing data-dependent preconditioner (multi-dimensional generalization of learning rate). Their empirical success is accompanied with provable guarantees: in any optimization trajectory with given gradients, the adapting preconditioner is comparable to the best in hindsight, in terms of rate of convergence to local optimality.

Their success has been a source of intense investigations over the past decade, since their introduction, with literature spanning thousands of publications, some highlights are surveyed below. The common intuitive understanding of their success is their ability to change the preconditioner, or learning rate matrix, per coordinate and on the fly. A methodological way of changing the learning rate allows treating important coordinates differently as opposed to commonly appearing features of the data, and thus achieve faster convergence.

In this paper we investigate whether a more refined goal can be obtained: namely, can we adapt the learning rate per coordinate, and also in short time intervals? The intuition guiding this search is the rising popularity in "exotic learning rate schedules" for training deep neural networks. The hope is that an adaptive learning rate algorithm can automatically tune its preconditioner, on a per-coordinate and per-time basis, such to guarantee optimal behavior even locally.

To pursue this goal, we use and improve upon techniques from the literature on adaptive regret in online learning to create a provable method that is capable of attaining optimal regret in any subinterval of the optimization trajectory. We then test the resulting method and compare it to learning a learning rate schedule from scratch.

## 1.1 Statement of our results

The (stochastic/sub)-gradient descent algorithm is given by the following iterative update rule:

$$x_{\tau+1} = x_\tau - \eta_\tau \nabla_\tau.$$

If $\eta_\tau$ is a matrix, it is usually called a preconditioner. A notable example for a preconditioner is when $\eta_\tau$ is equal to the inverse Hessian (or second differential), which gives Newton's method. Let $\nabla_1, ..., \nabla_T$ be the gradients observed in an optimization trajectory, the AdaGrad algorithm (and more general adaptive gradient methods) achieves the following convergence guarantee for convex optimization:

$$\sim \frac{\sqrt{\min_{H \in \mathcal{H}} \sum_\tau \|\nabla_\tau\|_H^{*2}}}{T},$$

where $\mathcal{H}$ is a family of matrix norms, most commonly those with a bounded trace. In this paper we improve upon this guarantee in terms of the local performance over any sub-interval of the optimization trajectory:

**Theorem 1** (Informal). *The convergence rate of Algorithm 1 can be upper bounded by:*

$$\tilde{O} \left( \frac{\min_k \min_{H_1,...,H_k \in \mathcal{H}} \sum_{j=1}^k \sqrt{\sum_{\tau \in I_j} \|\nabla_\tau\|_{H_j}^{*2}}}{T} \right)$$

The convergence result above is derived using the methodology of regret in online convex optimization (OCO). Our main technical contribution is a variant of the multiplicative weight algorithm, that achieves full-matrix regret bound over any interval by automatically selecting the optimal $\eta$. Previous methods came short of achieving this bound since the optimal $\eta$ depends on future gradients and cannot be determined in advance. Our algorithm achieves $\tilde{O}(\min_{H \in \mathcal{H}} \sqrt{\sum_{\tau=s}^t \nabla_\tau^\top H^{-1} \nabla_\tau})$ regret over all intervals simultaneously. A comparison of our results in terms of adaptive regret is given in Table 1.

| Algorithm | Regret over $I = [s, t]$ |
|---|---|
| [17] | $\tilde{O}(\sqrt{T})$ |
| [10], [22] | $\tilde{O}(\sqrt{|I|})$ |
| [9] | $\tilde{O}(\sqrt{\sum_{\tau=s}^t \|\nabla_\tau\|^2})$ |
| SAMUEL (ours) | $\tilde{O}(\sqrt{\sum_{\tau=s}^t \|\nabla_\tau\|_H^{*2}})$ |

Table 1: Comparison of results. We evaluate the regret performance of the algorithms on any interval $I = [s, t]$. For the ease of presentation we hide secondary parameters. Our algorithm achieves the regret bound of AdaGrad, which is known to be tight in general, but on any interval.

## 1.2 Related Work

Our work lies in the intersection of two related areas: adaptive gradient methods for continuous optimization, and adaptive regret algorithms for regret minimization, surveyed below.

**Adaptive Gradient Methods.** Adaptive gradient methods and the AdaGrad algorithm were proposed in [12]. Soon afterwards followed other popular algorithms, most notable amongst them are Adam [23], RMSprop [36], and AdaDelta [41].

Numerous efforts were made to improve upon these adaptive gradient methods in terms of parallelization, memory consumption and computational efficiency of batch sizes [32, 2, 15, 8].

A multitude of rigorous analyses of AdaGrad, Adam and other adaptive methods have appeared in recent literature, notably [38, 24, 11]. However, fully understanding the theory and utility of adaptive methods remains an active research area, with diverse (and sometimes clashing) philosophies [39, 31, 1].

[6] used the multiplicative weights update method for training deep neural networks that is more robust to learning rates than vanilla adaptive gradient methods.

**Adaptive Regret Minimization in Online Convex Optimization.** The concept of competing with a changing comparator was pioneered in the work of [20, 7] on tracking the best expert.

Motivated by computational considerations for convex optimization, the notion of adaptive regret was first introduced by [17], which generalizes regret by considering the regret of every interval. They also provided an algorithm Follow-The-Leading-History which attains $\tilde{O}(\sqrt{T})$ adaptive regret. [10] considered the worst regret performance among all intervals with the same length and obtain interval-length dependent bounds. [10] obtained an efficient algorithm that achieves $O(\sqrt{|I|\log^2 T})$ adaptive regret. This bound was later improved by [22] to $O(\sqrt{|I|\log T})$.

Recently, [9] improved previous results to a more refined second-order bound $\tilde{O}(\sqrt{\sum_{\tau \in I}\|\nabla_\tau\|^2})$, but in a more restricted setting assuming the loss is linear. These existing methods failed to achieve the optimal full-matrix rate, and we overcome this challenge by building a non-trivial variant of multiplicative weight algorithm which automatically chooses the optimal $\eta$.

For other related work, some considered the dynamic regret of strongly adaptive methods [45, 43]. [42] considered smooth losses and proposes SACS which achieves an $O(\sum_{\tau=s}^{t}\ell_\tau(x_\tau)\log^2 T)$ regret bound. There are also works utilizing strongly adaptive regret in online control [46, 30].

**Learning Rate Schedules and Hyperparameter Optimization.** On top of adaptive gradient methods, a plethora of nonstandard learning rate schedules have been proposed. The most commonly used one is the step learning rate schedule, which changes the learning rate at fixed time-points. A cosine annealing rate schedule was introduced by [27]. Alternative learning rates were studied in [3]. Learning rate schedules which increase the learning rate over time were proposed in [25]. Learning the learning rate schedule itself was studied in [40].

Related to our paper are general approaches for hyperparameter optimization (HPO), not limited to learning rate. In critical applications, researchers usually use a grid search over the entire parameter space, but that becomes quickly prohibitive as the number of hyperparameters grows. More sophisticated methods include gradient-based methods such as [29, 28, 13, 4] are applicable to continuous hyperparameters, but not to schedules which we consider. Bayesian optimization (BO) algorithms [5, 33, 35, 34, 14, 37, 21] tune hyperparameters by assuming a prior distribution of the loss function, and then keep updating this prior distribution based on the new observations.

## 2 Setting and Preliminaries

**Online convex optimization.** Consider the problem of online convex optimization (see [16] for a comprehensive treatment). At each round $\tau$, the learner outputs a point $x_\tau \in \mathcal{K}$ for some convex domain $\mathcal{K} \subset R^d$, then suffers a convex loss $\ell_\tau(x_\tau)$ which is chosen by the adversary. The learner also receives the sub-gradients $\nabla_\tau$ of $\ell_\tau()$ at $x_\tau$. The goal of the learner in OCO is to minimize regret, defined as

$$\text{Regret} = \sum_{\tau=1}^{T}\ell_\tau(x_\tau) - \min_{x \in \mathcal{K}}\sum_{\tau=1}^{T}\ell_\tau(x).$$

Henceforth we make the following basic assumptions for simplicity (these assumptions are known in the literature to be removable):

**Assumption 1.** *There exists $D, D_\infty > 1$ such that $\|x\|_2 \leq D$ and $\|x\|_\infty \leq D_\infty$ for any $x \in \mathcal{K}$.*

**Assumption 2.** *There exists $G > 1$ such that $\|\nabla_\tau\|_2 \leq G, \forall \tau \in [1, T]$.*

We make the notation of the norm $\|\nabla\|_H$, for any PSD matrix $H$ to be:

$$\|\nabla\|_H = \sqrt{\nabla^\top H \nabla}$$

And we define its dual norm to be $\|\nabla\|_H^* = \sqrt{\nabla^\top H^{-1}\nabla}$. In particular, we denote $\mathcal{H} = \{H | H \succeq 0, tr(H) \leq d\}$. An optimal blackbox online learning algorithm is also needed for our construction. We consider Adagrad from [12], which is able to achieve the following regret if run on $I = [s, t]$:

$$\text{Regret}(I) = O\left(Dd^{\frac{1}{2}}\min_{H \in \mathcal{H}}\sqrt{\sum_{\tau=s}^{t}\nabla_\tau^\top H^{-1}\nabla_\tau}\right)$$

93 **The multiplicative weight method.** The multiplicative weight algorithm is the method to achieve
94 vanishing regret in the prediction from expert advice problem. Similar to OCO, the regret is defined
95 as how much worse the accumulated loss is compared with the best expert. For example, the classical
96 Weighted Majority algorithm [26] achieves expected regret $O(\sqrt{T \log(N)})$ for binary prediction
97 with $N$ experts. The basic idea is to choose experts according to their weights, which are updated
98 each round by the performance of experts.

## 3 An Improved Adaptive Regret Algorithm

100 In this section, we give a variant of multiplicative weight algorithm 1, that given any black-box
101 OCO algorithm $\mathcal{A}$ as experts, achieves an $\tilde{O}\left(\sqrt{\min_{H \in \mathcal{H}} \sum_{\tau=s}^{t} \nabla_\tau^\top H^{-1} \nabla_\tau}\right)$ regret bound (w.r.t.
102 the experts) over any interval $J = [s, t]$ simultaneously. To be more specific, the total regret can be
103 written as $R_0(J) + R_1(J)$, where $R_0(J)$ is the regret of an expert $\mathcal{A}_J$ and $R_1(J)$ is the regret of the
104 multiplicative weight part for which we give the improved upper bound. The formal guarantee is the
105 following:

**Theorem 2.** *Under assumptions 1 and 2, the regret $R_1(J)$ of the multiplicative weight algorithm
part in Algorithm 1 satisfies that for any interval $J = [s, t]$,*

$$R_1(J) = O\left(D \log(T) \max\left\{G\sqrt{\log(T)}, d^{\frac{1}{2}} \sqrt{\min_{H \in \mathcal{H}} \sum_{\tau=s}^{t} \|\nabla_\tau\|_H^{*2}}\right\}\right)$$

106 In contrast, vanilla weighted majority algorithm achieves $\tilde{O}(\sqrt{T})$ regret only over the whole interval
107 $[1, T]$, and we improve upon the previous best result $\tilde{O}(\sqrt{t-s})$ [10] [22].

108 We introduce some definitions and notations needed in the algorithm. Without loss of generality, we
109 assume $T = 2^k$ and define the geometric covering intervals following [10]:

110 **Definition 3.** Define $S_i = \{[1, 2^i], [2^i + 1, 2^{i+1}], ..., [2^k - 2^i + 1, 2^k]\}$ for $0 \le i \le k$. Define
111 $S = \cup_i S_i$ and $S(\tau) = \{I \in S | \tau \subset I\}$.

112 For $2^k < T < 2^{k+1}$, one can similarly define $S_i = \{[1, 2^i], [2^i + 1, 2^{i+1}], ..., [2^i \lfloor \frac{T-1}{2^i} \rfloor + 1, T]\}$, see
113 [10]. Henceforth at any time $\tau$ the number of 'active' intervals is only $O(\log(T))$, this guarantees
114 that the running time and memory cost per round of SAMUEL is as fast as $O(\log(T))$.

115 It's worth to notice that $q$ doesn't affect the behavior of $\mathcal{A}_{I,q}$ and only takes affect in the multiplicative
116 weight algorithm, and that $r_\tau(I, q)$ and $x_\tau(I, q)$ doesn't depend on $q$ so we may write $r_\tau(I)$ and
117 $x_\tau(I)$ instead for simplicity.

118 Now we explain how our new technique overcomes the challenges we met. Previous methods failed
119 to achieve the optimal full-matrix bound, because it requires setting $\eta$ optimally in advance, however
120 the optimal value depends on future gradients which we can't anticipate.

121 The naive way to get an optimal $\eta$ is to run another meta MW algorithm to choose from different
122 $\eta$s (a similar idea was used in [44]), on top of any adaptive regret algorithm. However, though the
123 meta MW improves the regret of internal MWs, its own regret is sub-optimal again. Instead, we
124 incorporate the different $\eta$s to the experts of the internal MW.

125 We build our algorithm upon the framework of [10], but construct a set of candidate $\eta$ such that one of
126 them is guaranteed to be near-optimal, then make copies of each 'expert' $\mathcal{A}_I$ with different learning
127 rates $\eta$ in the multiplicative weight algorithm. The experts now no longer represent only different
128 intervals, but carry different $\eta$s as well. We prove Theorem 2 by first deriving an optimal full-matrix
129 regret bound on $R_1(I)$ for any $I \in S$. Then we use Cauchy-Schwarz to extend the regret bound to
130 any interval $J$.

131 **Remark 4.** The reason we can use convex combination instead in line 8 is because the loss $\ell_\tau$ and the
132 domain $\mathcal{K}$ are both convex. The convexity of $\mathcal{K}$ guarantees that $x_\tau$ still lies in $\mathcal{K}$, and the convexity of
133 $\ell_\tau$ guarantees that the loss suffered $\ell_\tau(x_\tau)$ is no larger than the expected loss of the random version:
134 $\sum_{I \in S(\tau), q} w_\tau(I, q) \ell_\tau(x_\tau(I, q)) / W_\tau$.

**Algorithm 1** Strongly Adaptive regret MUltiplicative-wEights (SAMUEL )

---

Input: OCO algorithm $\mathcal{A}$, geometric interval set $S$, constant $Q = 4\log(dTD^2G^2)$.
Initialize: for each $I \in S$, $Q$ copies of OCO algorithm $\mathcal{A}_{I,q}$.
Set $\eta_{I,q} = \frac{1}{2GD2^q}$ for $q \in [1, Q]$.
Initialize $w_1(I, q) = \min\{1/2, \eta_{I,q}\}$ if $I = [1, s]$, and $w_1(I, q) = 0$ otherwise for each $I \in S$.
**for** $\tau = 1, \dots, T$ **do**
    Let $x_\tau(I, q) = \mathcal{A}_I(\tau)$
    Let $W_\tau = \sum_{I \in S(\tau),q} w_\tau(I, q)$.
    Let $x_\tau = \sum_{I \in S(\tau),q} w_\tau(I, q)x_\tau(I, q)/W_\tau$.
    Predict $x_\tau$.
    Receive loss $\ell_\tau(x_\tau)$, define $r_\tau(I) = \ell_\tau(x_\tau) - \ell_\tau(x_\tau(I, q))$.
    For each $I = [s, t] \in S$, update $w_{\tau+1}(I, q)$ as follows,

$$w_{\tau+1}^{(I,q)} = \begin{cases} 0 & \tau + 1 \notin I \\ \min\{1/2, \eta_{I,q}\} & \tau + 1 = s \\ w_\tau(I, q)(1 + \eta_{I,q}r_\tau(I)) & \textbf{else} \end{cases}$$

**end for**

---

### 3.1 Proof of Theorem 2

*Proof.* We define the pseudo weight $\tilde{w}_\tau(I, q) = w_\tau(I, q)/\eta_{I,q}$ for $\tau \leq t$, and for $\tau > t$ we just set $\tilde{w}_\tau(I, q) = \tilde{w}_t(I, q)$. Let $\tilde{W}_\tau = \sum_{I \in S(\tau),q} \tilde{w}_\tau(I, q)$, we are going to show the following inequality

$$\tilde{W}_\tau \leq \tau(\log(\tau) + 1)\log(dTD^2G^2)\log(T) \tag{1}$$

We prove this by induction. For $\tau = 1$ it follows since on any interval $[1, t]$ the number of experts is exactly the number of possible $q$s, and the number of intervals $[1, t] \subset S$ is $O(\log(T))$. Now we assume it holds for all $\tau' \leq \tau$. We have

$$\tilde{W}_{\tau+1} = \sum_{I \in S(\tau+1),q} \tilde{w}_{\tau+1}(I, q)$$

$$= \sum_{I=[\tau+1,t]\in S(\tau+1),q} \tilde{w}_{\tau+1}(I, q) + \sum_{I=[s,t],s\leq\tau\in S(\tau+1),q} \tilde{w}_{\tau+1}(I, q)$$

$$\leq \log(\tau+1)\log(dTD^2G^2)\log(T) + 1 + \sum_{I=[s,t],s\leq\tau\in S(\tau+1),q} \tilde{w}_{\tau+1}(I, q)$$

$$= \log(\tau+1)\log(dTD^2G^2)\log(T) + 1 + \sum_{I=[s,t],s\leq\tau\in S(\tau+1),q} \tilde{w}_\tau(I, q)(1 + \eta_{I,q}r_\tau(I))$$

$$\leq \log(\tau+1)\log(dTD^2G^2)\log(T) + 1 + \tilde{W}_\tau + \sum_{I\in S(\tau),q} w_\tau(I, q)r_\tau(I)$$

$$\leq (\tau+1)(\log(\tau+1) + 1)\log(dTD^2G^2)\log(T) + \sum_{I\in S_\tau,q} w_\tau(I, q)r_\tau(I)$$

We further show that $\sum_{I\in S(\tau),q} w_\tau(I, q)r_\tau(I) \leq 0$:

$$\sum_{I\in S(\tau),q} w_\tau(I, q)r_\tau(I) = W_\tau \sum_{I\in S(\tau),q} p_\tau(I, q)(\ell_\tau(x_\tau) - \ell_\tau(x_\tau(I, q)))$$

$$\leq W_\tau \sum_{I\in S(\tau),q} p_\tau(I, q)\left(\sum_{J\in S(\tau),q} w_\tau(J, q)\ell_\tau(x_\tau(J, q))/W_\tau - \ell_\tau(x_\tau(I, q))\right)$$

$$= 0$$

which finishes the proof of induction.

Based on this, we proceed to prove that for any $I = [s, t] \in S$,

$$\sum_{\tau=s}^{t} r_\tau(I) = O\left(\sqrt{\log(T)} \max\left\{DG\sqrt{\log(T)}, \sqrt{\sum_{\tau=s}^{t}(\nabla_\tau^\top(x_\tau - x_\tau(I)))^2}\right\}\right)$$

By inequality 1, we have that

$$\tilde{w}_{t+1}(I, q) \le \tilde{W}_{t+1} \le (t+1)(\log(t+1)+1)\log(dTD^2G^2)\log(T)$$

Taking the logarithm of both sides, we have

$$\log(\tilde{w}_{t+1}(I, q)) \le \log(t+1) + \log(\log(t+1)+1) + \log(\log(dTD^2G^2)) + \log(\log(T))$$

Recall the expression

$$\tilde{w}_{t+1}(I, q) = \prod_{\tau=s}^{t}(1 + \eta_{I,q}r_\tau(I))$$

By using the fact that $\log(1+x) \ge x - x^2, \forall x \ge -1/2$ and

$$|\eta_{I,q}r_\tau(I)| \le \frac{1}{4GD}\|x_\tau - x_\tau(I, q)\|_2 G \le 1/2$$

we obtain for any $q$

$$\log(\tilde{w}_{t+1}(I, q)) \ge \sum_{\tau=s}^{t}\eta_{I,q}r_\tau(I) - \sum_{\tau=s}^{t}\eta_{I,q}^2 r_\tau(I)^2$$

Now we upper bound the term $\sum_{\tau=s}^{t}r_\tau(I)^2$. By convexity we have that $r_\tau(I) = \ell_\tau(x_\tau) - \ell_\tau(x_\tau(I)) \le \nabla_\tau^\top(x_\tau - x_\tau(I))$, hence

$$\sum_{\tau=s}^{t}r_\tau(I) \le \frac{4\log(T)}{\eta_{I,q}} + 4\eta_{I,q}\sum_{\tau=s}^{t}(\nabla_\tau^\top(x_\tau - x_\tau(I)))^2$$

The next step is to upper bound the term $\nabla_\tau^\top(x_\tau - x_\tau(I))$. By Hölder's inequality we have that $\nabla_\tau^\top(x_\tau - x_\tau(I)) \le \|\nabla_\tau\|_{H^{-1}}\|x_\tau - x_\tau(I)\|_H$ for any $H$. As a result, we have that for any $H$ which is PSD and $tr(H) \le d$,

$$(\nabla_\tau^\top(x_\tau - x_\tau(I)))^2 \le \nabla_\tau^\top H^{-1}\nabla_\tau\|x_\tau - x_\tau(I)\|_H^2 \le \nabla_\tau^\top H^{-1}\nabla_\tau 4D^2 d$$

where $\|x_\tau - x_\tau(I)\|_H^2 \le 4D^2 d$ is by elementary algebra: let $H = V^{-1}MV$ be its diagonal decomposition where $B$ is a standard orthogonal matrix and $M$ is diagonal. Then

$$\begin{aligned}\|x_\tau - x_\tau(I)\|_H^2 &= (x_\tau - x_\tau(I))^\top H(x_\tau - x_\tau(I))\\&= (V(x_\tau - x_\tau(I)))^\top MV(x_\tau - x_\tau(I))\\&\le (V(x_\tau - x_\tau(I)))^\top dIV(x_\tau - x_\tau(I))\\&\le 4D^2 d\end{aligned}$$

Hence

$$\sum_{\tau=s}^{t}r_\tau(I) \le \frac{4\log(T)}{\eta_{I,q}} + 4\eta_{I,q}D^2 d\min_{H}\sum_{\tau=s}^{t}\nabla_\tau^\top H^{-1}\nabla_\tau$$

The optimal choice of $\eta$ is of course

$$4\sqrt{\frac{\log(T)}{D^2 d\min_H \sum_{\tau=s}^{t}\nabla_\tau^\top H^{-1}\nabla_\tau}}$$

When $D^2 d\min_H \sum_{\tau=s}^{t}\nabla_\tau^\top H^{-1}\nabla_\tau \le 64G^2 D^2\log(T)$, $\eta_{I,1}$ gives the bound $O(GD\log(T))$. When $D^2 d\min_H \sum_{\tau=s}^{t}\nabla_\tau^\top H^{-1}\nabla_\tau > 64G^2 D^2\log(T)$, there always exists $q$ such that $0.5\eta_{I,q} \le \eta \le 2\eta_{I,q}$ by the construction of $q$ so that the regret $R_1(I)$ is upper bounded by

$$O\left(D\sqrt{\log(T)}\max\left\{G\sqrt{\log(T)}, d^{\frac{1}{2}}\sqrt{\min_{H \in \mathcal{H}}\sum_{\tau=s}^{t}\nabla_\tau^\top H^{-1}\nabla_\tau}\right\}\right) \tag{2}$$

Now we have proven an optimal regret for any interval $I \in S$, it's left to extend the regret bound to any interval $J$. We show that by using Cauchy-Schwarz, we can achieve the goal at the cost of an additional $\sqrt{\log(T)}$ term. We need the following lemma from [10]:

**Lemma 5** (Lemma 5 in [10])**.** *For any interval $J$, there exists a set of intervals $S^J$ such that $S^J$ contains only disjoint intervals in $S$ whose union is exactly $J$, and $|S_J| = O(\log(T))$*

We now use Cauchy-Schwarz to bound the regret:

**Lemma 6.** *For any interval $J$ which can be written as the union of $n$ disjoint intervals $\cup_i I_i$, its regret $Regret(J)$ can be upper bounded by:*

$$Regret(J) \leq \sqrt{n \sum_{i=1}^{n} Regret(I_i)^2}$$

*Proof.* The regret over $J$ can be controlled by $Regret(J) \leq \sum_{i=1}^{n} Regret(I_i)$. By Cauchy-Schwarz we have that

$$(\sum_{i=1}^{n} Regret(I_i))^2 \leq n \sum_{i=1}^{n} Regret^2(I_i)$$

which concludes our proof. $\qquad\square$

We can now upper bound the regret $R_1(J)$ using Lemma 6, replacing $Regret$ by $R_1$ and $n$ by $|S_J| = O(\log(T))$. For any interval $J$, its regret $R_1(J)$ can be upper bounded by:

$$R_1(J) \leq \sqrt{|S_J| \sum_{I \in S_J} R_1(I)^2}$$

Combining the above inequality with the upper bound on $R_1(I)$ 2, we reach the desired conclusion.
$\qquad\square$

## 3.2 Optimal Adaptive Regret with Adagrad Experts

In this subsection, we prove our main result as an application of Theorem 2, together with other extensions. Theorem 2 bounds the regret $R_1$ of the multiplicative weight part, while the total regret is $R_0 + R_1$. To get the optimal total regret bound, we only need to find an expert algorithm that also haves the optimal full-matrix regret bound matching that of $R_1$. As a result, we choose Adagrad as our expert algorithm $\mathcal{A}$, and prove regret bounds for both full-matrix and diagonal-matrix versions.

**Full-matrix adaptive regularization**   Our main result of this paper can be derived as a corollary from Theorem 2.

**Corollary 7** (Main Result)**.** *Under assumptions 1 and 2, when Adagrad is used as the blackbox $\mathcal{A}$, the total regret $Regret(I)$ of the multiplicative weight algorithm in Algorithm 1 satisfies that for any interval $I = [s, t]$,*

$$Regret(I) = O\left( D\log(T) \max\left\{ G\sqrt{\log(T)}, d^{\frac{1}{2}} \sqrt{\min_{H \in \mathcal{H}} \sum_{\tau=s}^{t} \|\nabla_\tau\|_H^{*2}} \right\} \right)$$

**Remark 8.** We notice that the $\log(T)$ overhead is brought by the use of $S$ and Cauchy-Schwarz. We remark here that by replacing $S$ with the set of all sub-intervals, we can achieve an improved bound with only a $\sqrt{\log(T)}$ overhead using the same analysis. On the other hand, such improvement in regret bound is at the cost of efficiency, that each round we need to make $\Theta(T)$ computations.

**Diagonal-matrix adaptive regularization**   If we restrict our expert optimization algorithm to be diagonal Adagrad, we can derive a similar guarantee for the adaptive regret.

**Corollary 9.** *Under assumptions 1 and 2, when diagonal Adagrad is used as the blackbox $\mathcal{A}$, the total regret $Regret(I)$ of the multiplicative weight algorithm in Algorithm 1 satisfies that for any interval $I = [s, t]$,*

$$Regret(I) = \tilde{O}\left( D_\infty \sum_{i=1}^{d} \|\nabla_{s:t,i}\|_2 \right)$$

Here $\nabla_{s:t,i}$ denotes the $ith$ coordinate of $\sum_{\tau=s}^{t} \nabla_\tau$.

## 4 Experiments

We demonstrate the effectiveness of our method on popular vision and language benchmarks: image classification on CIFAR-10 and ImageNet, and sentiment classification on SST-2. On all tasks, SAMUEL stably achieves high accuracy without learning rate schedule tuning.

For experiments, we made a few adjustments to our theoretical algorithm 1 to be computationally efficient in practice. We take a fixed number of experts with exponential decay factor on the history as shown below. Additionally, we sample experts instead of taking convex combination of them. In the original algorithm every expert's state is initialized once it becomes active, now that we don't have 'hard intervals' any longer, we change it to reinitialize all experts at fixed time-points. The below equation is the update rule of the Adagrad variant which we use for experiments. We use a parameter $\alpha$ to represent the memory length, which can be seen as a 'soft' version of Algorithm 1.

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

## 5 Conclusion

In this paper we study adaptive gradient methods with local guarantees. The methodology is based on adaptive online learning, in which we contribute a novel twist on the multiplicative weight method that we show has better adaptive regret guarantees than state of the art. This, combined with known results in adaptive gradient methods, gives an algorithm SAMUEL with optimal full-matrix local adaptive regret guarantees. We demonstrate the effectiveness and robustness of SAMUEL in experiments, where we show that SAMUEL can automatically adapt to the optimal learning rate and achieve comparable task accuracy as a fine-tuned optimizer, in a single run. While these experiments do not show improvement in state-of-the-art, they show potential of local adaptive gradient methods to be more robust to hyperparameter tuning.

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

# A   Appendix

# B   Proof of Corollary 7

*Proof.* Using Theorem 2 we have that $R_1(I)$ is upper bounded by

$$R_1(I) = O\left( D\log(T) \max\left\{ G\sqrt{\log(T)}, d^{\frac{1}{2}} \sqrt{\min_{H\in\mathcal{H}} \sum_{\tau=s}^{t} \|\nabla_\tau\|_H^{*2}} \right\} \right)$$

Because on each interval $J \in S$, one of the Adagrad experts achieve the bound

$$R_0(J) = O\left( Dd^{\frac{1}{2}} \sqrt{\min_{H\in\mathcal{H}} \sum_{\tau=s}^{t} \|\nabla_\tau\|_H^{*2}} \right)$$

For any interval $I$, using the result from [10] (Lemma 5) and Lemma 6 by replacing $Regret$ by $R_0$, it follows

$$R_0(I) = O\left( D\sqrt{\log(T)} d^{\frac{1}{2}} \sqrt{\min_{H\in\mathcal{H}} \sum_{\tau=s}^{t} \|\nabla_\tau\|_H^{*2}} \right)$$

397 Combining both bounds give the desired bound on $Regret(I)$.     □

# C   Proof of Corollary 9

*Proof.* The proof is almost identical to that of the previous corollary, observing that the regret $R_0(I)$ is $\tilde{O}(D_\infty \sum_{i=1}^{d} \|\nabla_{s:t,i}\|_2)$ due to [12], and the regret $R_1(I)$ remains $\tilde{O}(D\sqrt{\min_{H\in\mathcal{H}} \sum_{\tau=s}^{t} \nabla_\tau^\top H^{-1}\nabla_\tau})$, which is upper bounded by $\tilde{O}(D_\infty \sum_{i=1}^{d} \|\nabla_{s:t,i}\|_2)$.     □

# D   Deriving Local Optima from Regret

403 Though our theory so far is mostly for the convex setting, most practical optimization problems have
404 non-convex loss functions, and it's important to derive convergence guarantees for the non-convex
405 setting as well. The goal is now to find an approximate first order stationary point $x_\tau$ with small

---

**Algorithm 2** Finding Stationary Point with SAMUEL

---

Input: non-convex loss function $\ell$, horizon $T$, $\lambda \geq \frac{\beta}{2}$.
**for** $\tau = 1, \ldots, T$ **do**
    Let $\ell_\tau(x) = \ell(x) + \lambda\|x - x_\tau\|_2^2$.
    Update $x_{\tau+1}$ to be the output of Algorithm 1 with $\mathcal{A}$ to be Adagrad, starting at $x_\tau$, for $w_\tau$ steps.
**end for**

---

$\|\nabla_\tau\|_2$. In this section, we give a brief discussion on how to reduce the convergence rate of finding a first-order stationary point of a non-convex function $\ell$, to the regret bound of $\ell$.

In a nutshell, we adopt a method like GGT in [2] which is a proximal-point like algorithm that solves a sequence of convex sub-problems and guarantees to output an approximate stationary point. We assume that $\ell(x)$ is $\beta$-smooth and $\ell(x_1) - \min_x \ell(x) \leq M$. The use of Algorithm 1 can accelerate the convergence of each sub-problem, i.e. making $w_\tau$ smaller. The following proposition is direct from Theorem 1.

**Proposition 10.** $\ell_\tau(x_{\tau+1}) - \min_x \ell_\tau(x) =$

$$\tilde{O}\left(\frac{\min_k \min_{H_1,\ldots,H_k \in \mathcal{H}} \sum_{j=1}^k \sqrt{\sum_{\tau \in I_j} \|\nabla_\tau\|_{H_j}^{*2}}}{w_\tau}\right)$$

And we define the adaptive ratio $\mu(w_\tau)$ to be

$$\mu(w_\tau) = \frac{\min_k \min_{H_1,\ldots,H_k \in \mathcal{H}} \sum_{j=1}^k \sqrt{\sum_{\tau \in I_j} \|\nabla_\tau\|_{H_j}^{*2}}}{\sqrt{w_\tau(\ell(x_0) - \min_x \ell(x))}}$$

which quantifies the improvement of our adaptive algorithm by its advantage over the usual worst-case bound of vanilla SGD/Adagrad in $w_\tau$ rounds, see [2] for more details whose proof idea we follow. We are now ready to analyze the convergence rate of Algorithm 2. We begin by proving the following useful property for any $\eta > 0$:

$$\ell_\tau(x_\tau) - \min_x \ell_\tau(x) \geq \ell(x_\tau) - \ell_\tau(x_\tau - \eta\nabla_\tau)$$

$$\geq \eta\|\nabla_\tau\|_2^2 - \frac{\beta\eta^2}{2}\|\nabla_\tau\|_2^2 - \lambda\eta^2\|\nabla_\tau\|_2^2$$

Setting $\eta = \frac{1}{\beta+2\lambda}$, we have that

$$\ell_\tau(x_\tau) - \min_x \ell_\tau(x) \geq \frac{\|\nabla_\tau\|_2^2}{2(\beta + 2\lambda)} \tag{3}$$

Meanwhile, we have the following bound

$$\ell(x_\tau) - \ell(x_{\tau+1}) \geq \ell_\tau(x_\tau) - \ell_\tau(x_{\tau+1})$$
$$= \ell_\tau(x_\tau) - \min_x \ell_\tau(x) - (\ell_\tau(x_{\tau+1}) - \min_x \ell_\tau(x))$$
$$\geq \ell_\tau(x_\tau) - \min_x \ell_\tau(x) - \mu(w_\tau)\sqrt{\frac{\ell_\tau(x_\tau) - \min_x \ell_\tau(x)}{w_\tau}}$$

Fix $\epsilon > 0$, denote $w_\tau(\epsilon)$ to be the smallest integer that makes

$$\frac{\min_k \min_{H_1,\ldots,H_k \in \mathcal{H}} \sum_{j=1}^k \sqrt{\sum_{\tau \in I_j} \|\nabla_\tau\|_{H_j}^{*2}}}{w_\tau(\epsilon)(\ell(x_0) - \min_x \ell(x))} \leq \sqrt{\frac{\epsilon^2}{8(\beta + 2\lambda)}}$$

Suppose for contradiction now, that for all $\tau$, $\|\nabla_\tau\|_2 > \epsilon$, then $\ell(x_\tau) - \ell(x_{\tau+1}) \geq \frac{\ell_\tau(x_\tau)-\min_x \ell_\tau(x)}{2} \geq \frac{\|\nabla_\tau\|_2^2}{4(\beta+2\lambda)}$ by property 3 and the definition of $w_\tau(\epsilon)$. Summing over $[1, T]$

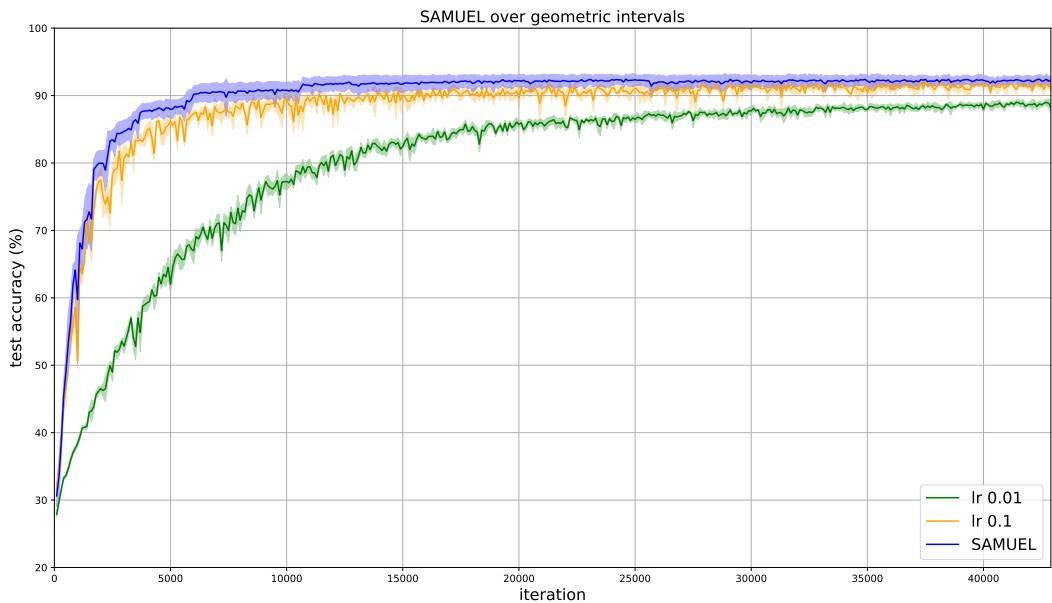

Figure 3: SAMUEL over geometric intervals on CIFAR-10.

we get

$$\ell(x_1) - \ell(x_{T+1}) \geq \frac{T\epsilon^2}{4(\beta + 2\lambda)}$$

If we set $T = \frac{4M(\beta+2\lambda)}{\epsilon^2}$, then the above inequality will lead to contradiction. Therefore, within $\sum_{\tau=1}^{T} w_\tau(\epsilon)$ calls of Algorithm 2, it's guaranteed that our algorithm will output some $x_\tau$ that $\|\nabla_\tau\| \leq \epsilon$. We can rewrite the number of calls in terms of the adaptive ratio: $O(\frac{\overline{\mu(w_\tau(\epsilon))^2}}{\epsilon^4})$, concerning only $\epsilon$ and letting $\overline{\mu(w_\tau(\epsilon))}$ denote the average of all $\mu(w_\tau(\epsilon))$. Comparing with the convergence rate $O(\frac{1}{\epsilon^4})$ of SGD, we make improvement when the optimization trajectory is more adaptive.

**Theorem 11** (Informal). *The convergence rate of Algorithm 2, is $O(\frac{\overline{\mu(w_\tau(\epsilon))}^2}{\epsilon^4})$ ignoring parameters except $\epsilon$.*

# E   Additional Experiments

## E.1   Experiments with online switching

In this section we conduct a preliminary sanity check to test SAMUEL ability to switch learning rates on the fly. For this purpose we tested the full SAMUEL implementation with the original Algorithm 1 on CIFAR-10 classification. We compared training ResNet-18 with SAMUEL to training with AdaGrad with constant learning rate multiplier as shown in Fig3. For the baseline learning rate multiplier, we considered multiplier of 0.01 and 0.1. For SAMUEL, we constructed the geometric interval set with the minimum length of 100 training iterations and provided multipliers 0.01 and 0.1 as candidate learning rate multipliers to SAMUEL. Although SAMUEL can only alternate between two candidate learning rate multipliers, it demonstrates superior performance. Baselines and SAMUEL over geometric intervals were both trained for 220 epochs with batch size of 256. We conducted experiments with 5 different random seeds for each of three schedules 0.01, 0.1 and SAMUEL . We report the average final test accuracy: 88.98% with lr 0.01, 92.08% with lr 0.1, and 92.43% with SAMUEL .

In this experiment SAMUEL prefers lr 0.1 at first, then switch to lr 0.01 automatically around iteration 2500, where it starts to outperform the lr 0.1 baseline. It demonstrates the ability of SAMUEL to switch between learning rates on the fly.

This shows the promise of interpolating different algorithms in a manner that improves upon the individual methods. However, this implementation not as efficient as the heuristic we test in the other experiments.

It remains to test how quickly we can shift optimizers in more challenging online tasks, such as domain shift and online reinforcement learning.

## E.2 CIFAR-100 Experiment

We conducted image classification on the CIFAR-100 dataset. We compare a ResNet-18 [18] model trained with our optimization algorithm to a model trained with AdaGrad using brute-force searched learning rate schedulers. Following [18], we applied per-pixel mean subtraction, horizontal random flip, and random cropping with 4 pixel padding for CIFAR data processing and augmentation. All experiments were conducted on TPU-V2 hardware. For training, we used a batch size of 256 and 250 total epochs with a step learning rate schedule. We fixed the learning rate stepping point at epoch 125 and 200, and provided five possible candidate learning rates {0.0001, 0.001, 0.01, 0.1, 1} for each region. Thus an exhaustive search yielded 125 different schedules for the baseline AdaGrad method. For a fair comparison, we adopted the same learning rate changing point for our method. Our method automatically determined the optimal learning rate at the transition point without the need to exhaustively search over learning rate schedules.

We display the CIFAR-100 test accuracy curves of AdaGrad with 125 exhaustively-searched learning rate schedules and our method in only one single run in Fig.4. Fig.4 shows that the best accuracy of exhaustive search is 76.77%, and the accuracy of SAMUEL using the same seed is 75.66%.

## E.3 Comparison with Baselines

We conducted additional experiments on CIFAR-10 with off-the-shelf learning rate schedulers from the optax library. We considered the same model and training pipeline as detailed in the experiment section. Instead of using the three phase learning rate stepping scheme, we tried more varieties of schedulers available in the optax library. Specifically, we finetuned the cosine annealing scheduler, the linear warmup followed by cosine decay scheduler, and the linear warmup followed by exponential decay scheduler. Their test accuracy curves together with different learning rate schedules are displayed in Fig.5, Fig.6 and Fig.7, respectively.

For finetuning the cosine annealing scheduler, we experimented with 45 different initial learning rates in the range of 1e-5 to 0.9.

For the linear warmup followed by cosine decay scheduler, we finetuned the initial learning rate, the peak learning of the warmup and the duration of the warmup. We considered possible initial learning rate $\{0, 1 \times 10^{-5}, 1 \times 10^{-4}\}$, peak learning rate {0.001, 0.01, 0.05, 0.1, 0.5, 1}, and warmup epochs {5, 10} for the grid search.

For the linear warmup followed by exponential decay scheduler, we finetuned the initial learning rate, the peak learning of the warmup and the duration of the warmup, the exponential decay rate, and the transition steps. We considered possible initial learning rate $\{0, 1 \times 10^{-5}, 1 \times 10^{-4}\}$, peak learning rate {0.05, 0.1, 0.5, 1}, warmup epochs {5, 10}, exponential decay rate {0.5, 0.8, 0.9}, and transition step {5, 10} for the grid search.

As the figures demonstrate, the final test accuracy depend heavily on the learning rate schedules. For off-the-shelf learning rate schedulers, tuning the schedule associated hyperparameters is not trivial.

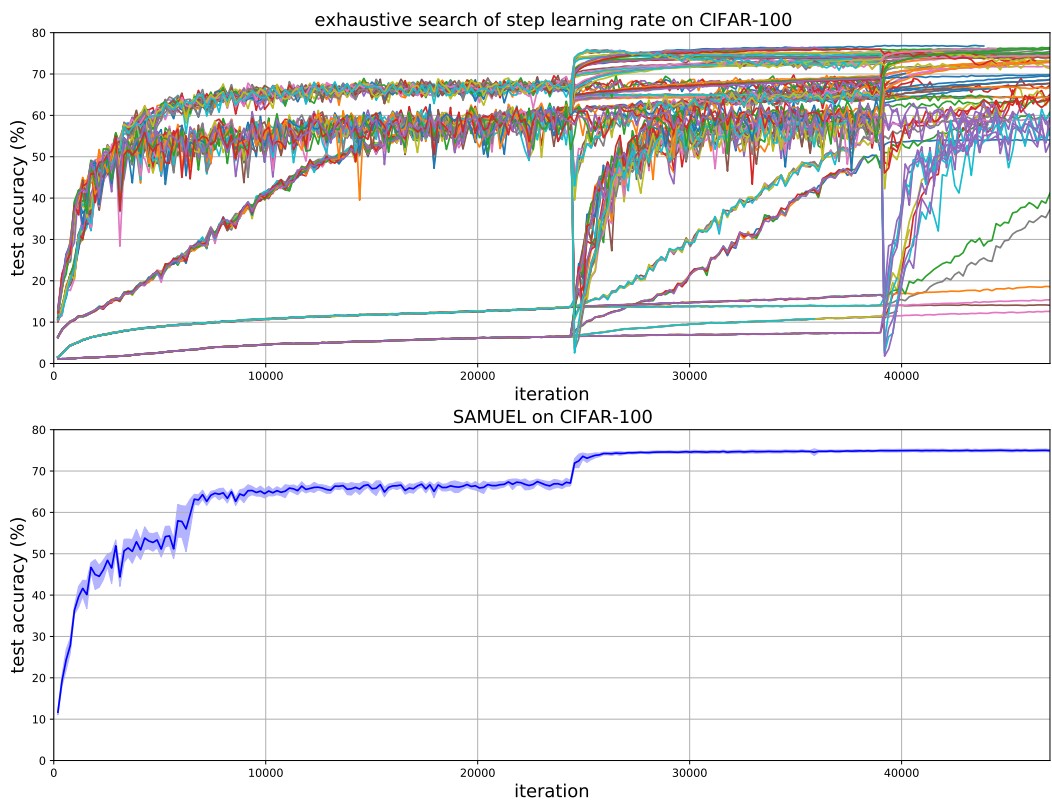

Figure 4: CIFAR-100 comparison of exhaustive searched learning rate schedule and SAMUEL . Top: 125 parallel experiments with exhaustively searched learning rate schedules. Bottom: SAMUEL on one run with 10 different random seeds, no tuning needed.

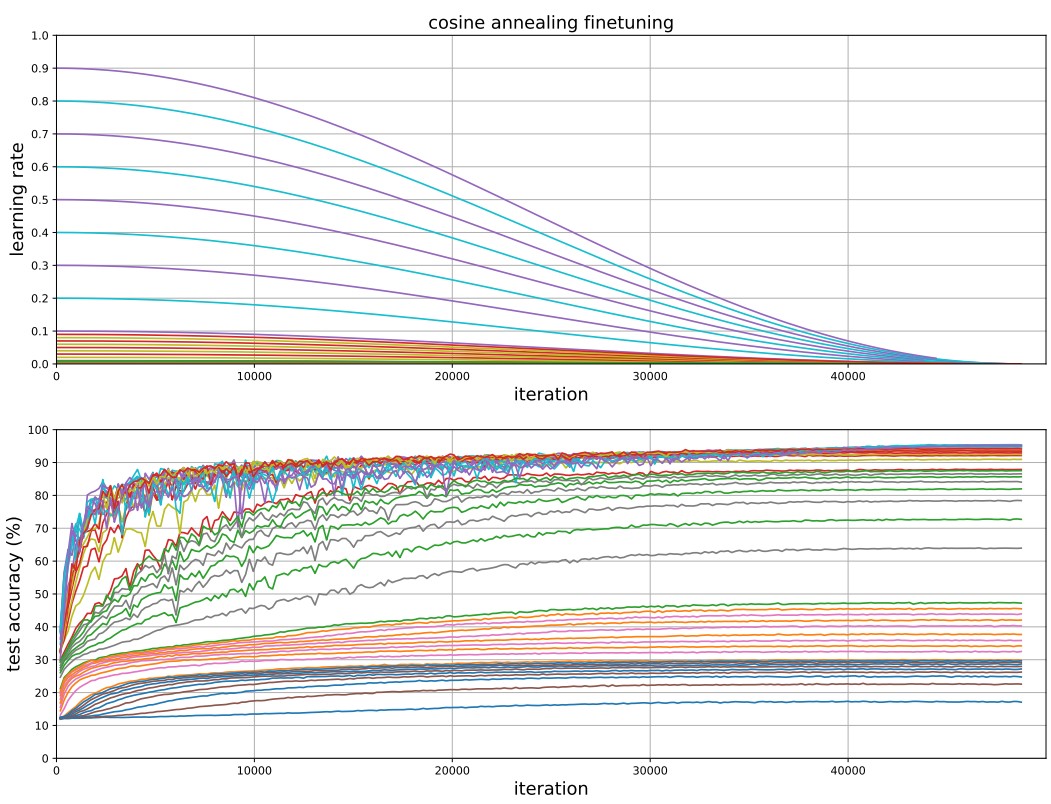

Figure 5: Tuning cosine annealing schedules on CIFAR-10. The best test accuracy out of all 45 trials is 95.37%.

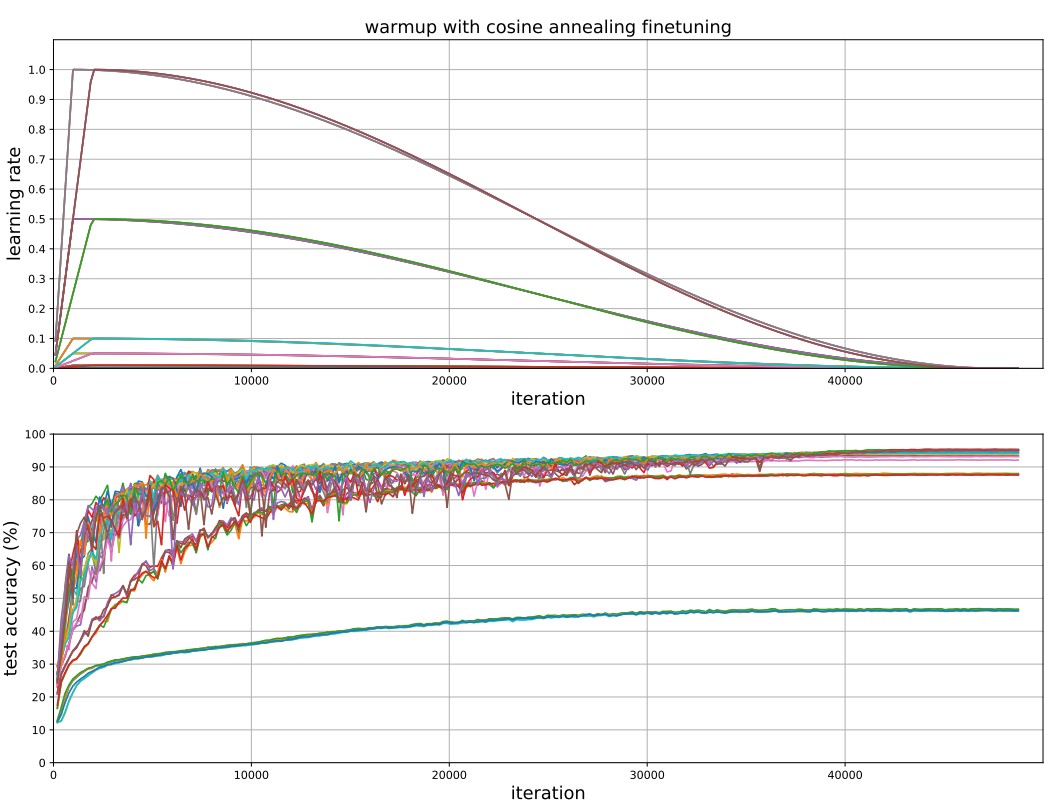

Figure 6: Tuning the linear warmup followed by cosine decay scheduler on CIFAR-10. The best test accuracy out of 36 trials is 95.31%.

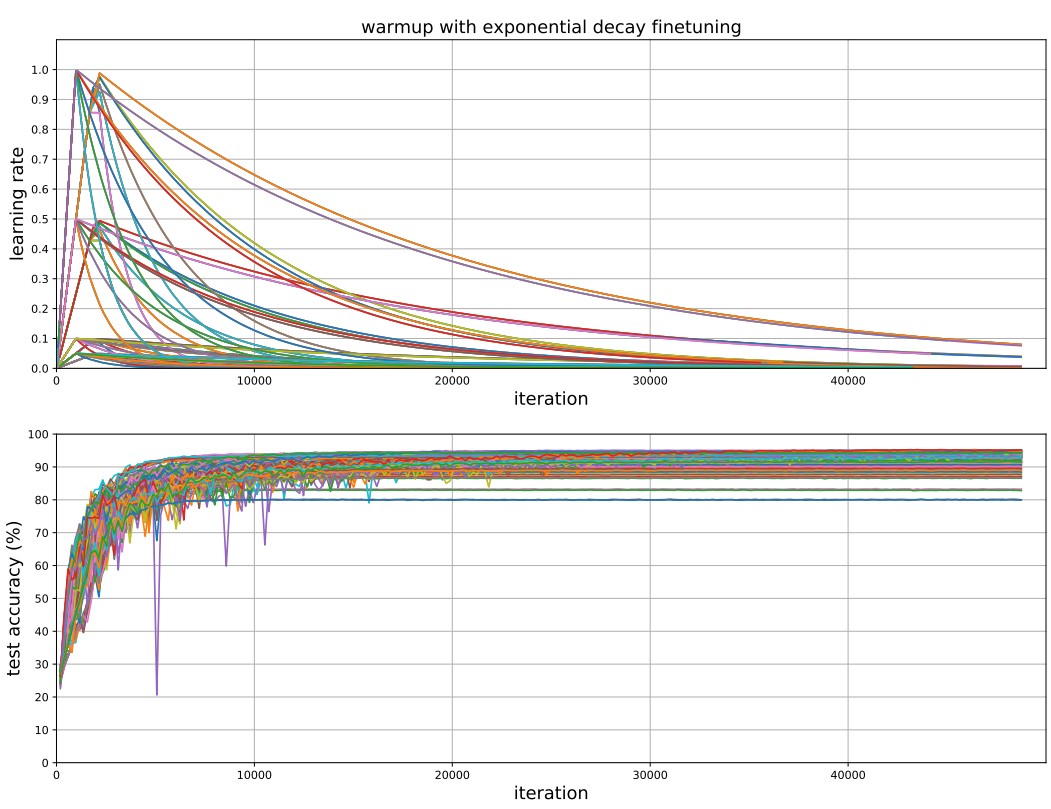

Figure 7: Tuning the linear warmup followed by exponential decay scheduler on CIFAR-10. The best test accuracy out of 144 trials is 95.27%.