# OpenReview forum: "Adaptive Gradient Methods with Local Guarantees"
_automl.cc/AutoML/2022/Workshop/Late_Breaking — AutoML 2022 (Late-Breaking Workshop)_

### Meta-Review · Area_Chair_fi1b · 2022-05-11

**Recommendation:** Accept
**Confidence:** 3

**Metareview:**

The paper should be improved according to reviewer suggestions, but is otherwise a good fit for the workshop and all reviewers find it interesting. PBT and its variations, as well as meta-gradient approaches seem to be relevant baselines for a future main-track submission.

Overall, it is a good fit for the workshop track and I happily recommend acceptance.

---

### Decision · Program_Chairs · 2022-05-13

Accept